# A Real-Life Study in Patients Newly Diagnosed with Autoimmune Hashimoto’s Thyroiditis: Analysis of Asthenia as Admission Complaint

**DOI:** 10.3390/life14111380

**Published:** 2024-10-27

**Authors:** Ana Valea, Mihai Costachescu, Mihaela Stanciu, Claudiu Nistor, Oana-Claudia Sima, Mara Carsote, Tiberiu Vasile Ioan Nistor, Denisa Tanasescu, Florina Ligia Popa, Mihai-Lucian Ciobica

**Affiliations:** 1Department of Endocrinology, “Iuliu Hatieganu” University of Medicine and Pharmacy, 400012 Cluj-Napoca, Romania; ana.valea@umfcluj.ro; 2Department of Endocrinology, County Emergency Clinical Hospital, 400347 Cluj-Napoca, Romania; 3PhD Doctoral School of “Carol Davila” University of Medicine and Pharmacy, 010825 Bucharest, Romania; mihaicostachescu@gmail.com; 4Thoracic Surgery Department, “Dr. Carol Davila” Central Emergency University Military Hospital, 010825 Bucharest, Romania; 5Department of Endocrinology, “Lucian Blaga” University of Sibiu, Victoriei Blvd., 550024 Sibiu, Romania; mihaela.stanciu@ulbsibiu.ro; 6Department of Endocrinology, Clinical County Emergency Hospital, 550245 Sibiu, Romania; 7Department 4—Cardio-Thoracic Pathology, Thoracic Surgery II Discipline, “Carol Davila” University of Medicine and Pharmacy, 050474 Bucharest, Romania; 8Department of Endocrinology, “Carol Davila” University of Medicine and Pharmacy, 050474 Bucharest, Romania; carsote_m@hotmail.com; 9Department of Clinical Endocrinology V, C.I. Parhon National Institute of Endocrinology, 011863 Bucharest, Romania; 10Medical Biochemistry Discipline, “Iuliu Hatieganu” University of Medicine and Pharmacy, 400347 Cluj-Napoca, Romania; tiberiu.nistor@umfcluj.ro; 11Medical Clinical Department, Faculty of Medicine, “Lucian Blaga” University of Sibiu, 550169 Sibiu, Romania; denisa.tanasescu@ulbsibiu.ro; 12Department of Physical Medicine and Rehabilitation, Faculty of Medicine, “Lucian Blaga” University of Sibiu, 550024 Sibiu, Romania; florina-ligia.popa@ulbsibiu.ro; 13Department of Internal Medicine and Gastroenterology, “Carol Davila” University of Medicine and Pharmacy, 020021 Bucharest, Romania; lucian.ciobica@umfcd.ro; 14Department of Internal Medicine I and Rheumatology, “Dr. Carol Davila” Central Military University Emergency Hospital, 010825 Bucharest, Romania

**Keywords:** antibody, antibody, autoimmune, asthenia, thyroid, thyroiditis, endocrine, hormone

## Abstract

**Background:** Amid the large panel of autoimmune thyroid diseases, Hashimoto’s thyroiditis (HT) represents a major point across multidisciplinary daily practice. When it comes to the clinical picture, particularly in regard to asthenia (also described as “fatigue” or “decreased energy”), the differential diagnosis is challenging, and a meticulous anamnesis should be backed up by focused lab investigations. Our objective was to analyze the thyroid panel in newly diagnosed patients with HT in relationship with the presence of asthenia as an admission complaint. **Methods:** This was a retrospective, multi-centric, real-life study conducted in secondary endocrine units (university hospitals) from July 2022 to July 2023. The exclusion criteria were COVID-19 infection; an active malignancy, etc. **Results:** The cohort (N = 120) included an asthenia group (AS, 49.2%) and a non-AS group of a similar age (49.3 ± 14.7 vs. 47.1 ± 14.8 y, *p* = 0.426). Headache was more frequent in the AS group (35.6% vs. 18%, *p* = 0.03). Thyroid function and HT-related antibodies assays were similar between the groups and show no correlation with serum total cholesterol and triglycerides, respectively. TSH levels did not vary among the age sub-groups (*p* = 0.701). One third of the studied population was affected by hypothyroidism (TSH > 4.5 μIU/mL), being seen at a higher rate in the AS (39%) vs. non-AS group (23%). Total cholesterol positively correlated with the patients’ age (r = 0.180, *p* = 0.049) and triglycerides (N = 120; r = 0.324, *p* < 0.001), as found only in the non-AS group (r = 0.246, *p* = 0.006, respectively, r = 0.319, *p* < 0.001). **Conclusions:** The analysis of the AS vs. non-AS group pinpointed the fact that, in regard to daily practice, asthenia as an admission complaint seems less of an indicator of an underlying thyroid dysfunction or a higher level of serum antibodies against thyroid in patients without a full clinical picture of thyrotoxicosis or myxoedema.

## 1. Introduction

Amid the large panel of autoimmune thyroid diseases, chronic Hashimoto’s thyroiditis (HT) represents a major point in daily practice from a multidisciplinary perspective due to a certain level of impaired quality of life when dealing with an associated thyroid dysfunction, as well as the co-presence of various endocrine and non-endocrine autoimmune comorbidities and other non-specific complaints, such as chronic fatigue/asthenia, sleep disturbances, and a tendency to gain weight or pain [1,2,3].

HT, the most common autoimmune thyroid ailment in the endocrine practice, may present with normal thyroid function, as well as hypothyroidism or even transitory (flare up) thyrotoxicosis. HT represents the most important cause of hypothyroidism in non-iodine deficient geographic areas, while approximately 30% to 60% of the HT subjects may be affected by hypothyroidism across their entire lifespan. A combination of genetic susceptibility, epigenetic, and/or environmental factors may be involved in developing an abnormal thyroid function in HT. Overall, the consequence of these contributors is the loss of immune tolerance and the exposure of the thyroid tissue to an autoimmune attack. This translates into lymphocytic findings in histological reports and follicular destruction in association with a thyroid gland atrophy and fibrosis. A large area of general and local symptoms has been reported in HT, but no specific clinical element is described in HT (other than highly suggestive features of hypothyroidism of any cause), while, generally, one fifth of the HT subjects are affected by other (non-HT) autoimmune disorders [1,2,3,4].

When it comes to the clinical picture, particularly in regard to asthenia (also described as “fatigue” or “decreased energy”), the differential diagnosis is challenging, and a meticulous anamnesis should be backed up by focused lab and imaging investigations [4,5,6]. However, asthenia should be potentially placed in relationship with the anomalies of the thyroid profile in any situation regardless of the positive traditional serum antibodies against the thyroid, such as anti-thyroperoxidase antibodies (TPOAbs) and anti-thyroglobulin antibodies (TgAbs) [7,8,9].

The recent COVID-19 pandemic highlighted many prior known or unknown medical and surgical entities and unexpected outcomes, including in the endocrine field [10,11,12]. One of the most interesting features of this issue was the post-COVID-19 asthenia (or fatigue) as a single issue or in addition to a complex picture that has been described as long COVID-19 syndrome, which sometimes involved transitory or permanent thyroid anomalies [13,14,15].

Our objective was to analyze the thyroid panel (thyroid function and HT-related antibodies assays) in patients newly diagnosed with HT in relationship with the presence of asthenia as an admission complaint. The importance of addressing this symptom comes from its high prevalence (including in HT) in daily multidisciplinary practice, while this clinical element (as a standalone input) harbors a rather low index of clinical suspicion, hence its importance as a diagnosis marker.

## 2. Materials and Methods

### 2.1. Study Design and Studied Population

This was a cross-sectional, retrospective, multi-centric, one-year, real-life study conducted in secondary endocrine units and associated departments (university hospitals) from July 2022 to July 2023.

The studied individuals were patients who had at least one admission, according to their hospitalization (inpatients) report, that included the assessment of a thyroid profile. The inclusion criteria were as follows: informed signed consent of the patient or at least one of the parents for subjects younger than 18 years old according to the each hospital protocol for day-by-day hospitalization (aged between 10 and 90 years); available data for the thyroid panel assessment as shown below; available data with concern to the medical records of prior co-morbidities (such as diabetes mellitus, anemia, cancers, COVID-19 infection, vaccination against coronavirus, and depression, etc.). All included patients had a confirmation of HT at current admission based on increased serum TPOAbs and/or TgAbs. The asthenia group included subjects who complained of asthenia (self-declared) during anamnesis for the previous one to three months (at least two days per week). This represented the AS group. The individuals who denied asthenia at admission were included in the non-AS group.

The exclusion criteria were as follows: a current infection or the diagnosis of an infection within last 6 months; the confirmation or suspicion of COVID-19 infection (including post-COVID-19 syndrome) at any point in time; current or prior (within one year) vaccination against coronavirus; an active malignancy or the diagnosis of a cancer at any point in time; any neurological ailment (either an acute neurological disease) within the previous 24 months to the current endocrine assessment or any confirmed neurologic chronic condition or depression; current or prior acute or chronic kidney disease as well as chronic respiratory or cardiovascular disorders. The specific endocrine elements of exclusion were the following: pregnancy (suspected or confirmed); clinical diagnosis of myxoedema, thyrotoxicosis, or subacute thyroiditis; post-thyroidectomy status; active functioning endocrine tumors; acute or chronic adrenal insufficiency (primary or secondary); prior diagnosis of HT; prior or concurrent therapy with levothyroxine or thiamazol; current or previous exposure to glucocorticoids (within 24 months); current diagnosis of anemia based on hemogram assays or prior diagnosis of a condition associated with anemia within the last 12 months; prior or current diagnosis of diabetes mellitus or high blood pressure of any type; current or prior (within 6 months) medication for dyslipidaemia. If a patient was suspected of having a thyroid malignancy amid the current admission/hospitalization, this subject was not included (regardless of the results of further investigations, such as a fine needle aspiration biopsy or post-thyroidectomy histological analysis).

The demographic features were registered (age, sex, rural/urban residence). As mentioned, asthenia was registered as a complaint during anamnesis (self-declared) if it was persistent for one to three months before the current admission (for at least two days per week). Also, the presence of a non-specific headache during the latest one to three months prior to admission was registered based on anamnesis (at least two episodes per week). The clinical exam included a general physical exam, thyroid palpation, and a body mass index calculation (kg/sqm). Obesity was considered based on a value of body mass index of between 30 and 34.9 kg/sqm (grade I), between 35 and 39.9 kg/sqm (grade II), and of at least 40 kg/sqm for grade III. The medical records were searched, and medical and surgical histories of autoimmune diseases (e.g., celiac disease, psoriasis, vitiligo, and urticaria), metabolic disorders (e.g., high blood pressure, and diabetes mellitus), and other medical issues (e.g., osteoporosis, cancers, etc.) were registered, including the ones that have been mentioned above. Blood assessments included examining the following information the thyroid panel: the Thyroid Stimulating Hormone (TSH), free levothyroxine (FT4), TPOAbs, TgAbs, as well as total cholesterol and triglycerides. All patients were tested for coronavirus infection on admission according to each hospital rule at that point of the COVID-19 pandemic. Newly diagnosed individuals with diabetes mellitus, anemia, or acute/chronic kidney disease were not included. Clinical suspicion of an adrenal insufficiency (of primary or secondary type) was selectively followed by baseline blood morning cortisol and Adrenocorticotropic Hormone (ACTH) assays; for a smaller number of patients, a Cosyntropin (ACTH) stimulation test was conducted based on the individual decision of the current physician. As mentioned, subjects with complete or partial adrenal failure were excluded. An anterior neck ultrasound was performed during hospitalization in each center. The data collection criteria included a homogenous/inhomogeneous pattern; echogenity (hypo-, iso- or hyperechoic); and the presence of at least one thyroid nodule of at least 5 mm diameter (this was quantified as “present” or “absent”). Active endocrine tumors were investigated based on the clinical individual decision and were excluded from the final analysis (Figure 1).

### 2.2. Statistical Analysis

The results were processed and analyzed using Excel 16.59 (Microsoft, Redmond, WA, USA) and SPSS 29.0.2.0 (SPSS, Inc., Chicago, IL, USA). The distribution of each numeric value was examined empirically and using normality tests, such as the Kolmogorov–Smirnov test. The central tendencies of Gaussian distributed values were expressed as mean ± standard deviation, and those not normally distributed were expressed as quartiles (Q1, Q2/median and Q3), with Q1 representing the first quartile (the value under which 25% of data points are found), while Q3 represented the third quartile (or upper quartile or 75th quartile). A Student’s t-test for the equality of means was used when comparing groups with continuous and normally distributed data, and a Mann–Whitney test was used for data without normal distribution. An analysis of variance test (ANOVA test) or Kruskal–Wallis test was used to compare the multiple groups. A chi-square test or Fisher’s exact test was used to test associations between groups with quantitative data. A univariate correlation between normal distributed values was tested by applying Pearson’s correlation coefficient. For continuous values with non-normal distributions, a Spearman’s correlation coefficient was also used. A null hypothesis was rejected at a *p*-value below 0.05.

### 2.3. Ethical Aspects

The patients signed the informed consent form at the moment of admission according to each hospital’s rules. The Local Ethics Committees approved the retrospective data collection. This analysis was part of the PRECES study (Parameters of Romanian population with endocrine conditions with or without endocrine surgery: real-world evidence and retrospective study), which is a multi-centric collaborative in the field of endocrinology and associated domains amid everyday practice in university hospitals. The PRECES study was approved by the Ethics Committees of the hospitals (number 124-25.06.2024, number 6284-08.02.2024; number 702-28.06.2024, and number 2058-30.01.2024).

## 3. Results

A total of 120 participants were included, with a number of 59 (49.2%) subjects in the AS group. Regarding age demographics (range between 10 and 81 years), the average age was similar, at 49.3 years in the AS group compared to 47.1 years in the non-AS group (*p* = 0.426). Rural residence involved a statistically significant higher number of patients in the AS versus non-AS group (42.3% versus 13.1%, *p* = 0.001). Autoimmune diseases (other than HT and adrenal insufficiency) affected between 10% and 14% of the subjects (*p* = 0.448). In terms of the non-autoimmune co-morbidities, out of the entire studied sample, 35.8% of the participants had obesity (of any grade) and 22.5% of them had osteoporosis (similarly between the two groups).

With respect to headache, as registered during anamnesis, the AS group was associated with a statistically significant higher rate compared to the non-AS group (35.6% versus 18.0%, *p* = 0.03) (Table 1).

A similar number of patients with either asthenia or headache was found among age groups (*p* = 0.431 and *p* = 0.843, respectively). The highest frequency of headache was self-reported in the 50–59 years group and then in the 40–49 years group, respectively, and there were similar results for asthenia with respect to the age group analysis. The following age groups showed a higher rate of asthenia than headache: 20–29 years, 40–49 years, 50–59 years, 60–69 years, and 70–79 years, respectively, while the 30–39 years group showed a similar rate (Figure 2).

Among the AS patients with obesity, 30.4% had grade I obesity, 39.1% had grade II, and 30.4% had grade III, respectively; for the non-AS group, these amounts were 65%, 30%, and 5% (*p* = 0.035) (Table 2, Figure A1).

One third of the studied population was affected by hypothyroidism, as defined by TSH levels above 4.5 μIU/mL, with a higher rate being found in the AS group (39%) versus non-AS group (23%) and in the age group between 40 and 49 years (40.7%) (Table 3).

TSH, FT4, TPOAbs and TgAbs were similar between the two groups, as were the total cholesterol and triglycerides levels (Table 4).

The entire cohort was divided by age in decades, and the variability of the TSH and FT4 mean was analyzed. The mean value of FT4 had an exponential ascendance beginning with the 60–69 years age group and reached the highest mean point in the oldest age group of 80–89 years (*p* = 0.043). TSH levels did not significantly vary among the age groups (*p* = 0.701) (Figure 3).

TPOAbs had a borderline significance by age groups in the AS group (*p* = 0.06) and did not statistically significant vary in the non-AS group (*p* = 0.602). In the AS group, the median of the TPOAbs had an exponential increase beginning in the 10–19 years age group, reached a peak median value in the 40–49 years age group, and then had a continuous decline (Figure 4).

When comparing the AS and non-AS group, TPOAbs had a similar distribution (*p* = 0.270) (Figure 5).

Parameters that belonged to the thyroid panel were compared with those of the biochemistry (lipid) profile, and a weak positive correlation between TgAb and triglycerides in the AS group (r = 0.185, *p* = 0.040) was found (Table 5).

Total cholesterol positively correlated with the patients’ age (N = 120; r = 0.180, *p* = 0.049) and triglycerides (N = 120; r = 0.324, p<0.001). Triglycerides correlated with the subjects’ age (r = 0.277, *p* = 0.002) in the total studied population. In the non-AS group, total cholesterol positively correlated with the patients’ age (r = 0.246, *p* = 0.006) and triglycerides (r = 0.319, *p* < 0.001), respectively (Table 6, Figure 6).

In the AS group, the TSH levels did not correlate with TPOAbs (r = 0.168, *p* = 0.061) (Figure 7).

Also, a positive correlation was found between TSHs and TPOAbs in the non-AS group (r = 0.215, *p* = 0.014) (Figure 8).

Neck ultrasonography showed the overall predominance of the inhomogeneous and hypoechoic pattern and the absence of thyroid nodules. Of the participants with AS, 94.9% had an inhomogeneous thyroid aspect, while 69.5% of them had a hypoechoic pattern and 74.6% did not display any thyroid nodules. The number of participants with a homogeneous, hypoechoic pattern and absent nodules was similar between both the studied sub-groups (Table 7).

## 4. Discussion

In this retrospective analysis, we collected the data on the thyroid profile in patients who were admitted for a thyroid check-up, noting that almost half of the studied HT population presented asthenia on first admission (AS group). In real-life settings, asthenia/fatigue represents a non-specific symptom that might require an endocrine evaluation if no other cause is obvious, such as the presence of severe oncologic, infectious, neurologic, etc. co-morbidities (as we specified in the exclusion criteria) [16,17,18,19,20]. In this instance, depending on each country protocols, these endocrine assays may be carried out not only amid those in primary health care but also in different units of multidisciplinary hospitals, as seen in this study [21].

Analyzing the TSH, FT4, TPOAb, and TgAb levels, we found a similar profile in the AS versus the non-AS group, therefore confirming that, for daily practice, asthenia as a complaint hardly serves as a good indicator of an underlying thyroid dysfunction or a higher level of serum antibodies against the thyroid, noting that the patients with a full clinical picture of thyrotoxicosis or myxedema were not included. Approximately one third of the studied population with newly diagnosed HT had TSH levels above the upper normal limit, with the highest rate being seen in the AS group. Other factors (e.g., subclinical conditions or co-morbidities) might explain why the study found no strong correlation between asthenia and thyroid dysfunction or antibody levels, and this might serve as an important aspect for practitioners. Most data from the literature showed the fact that the clinical picture, apart from a clear thyroid dysfunction, is not representative of the levels of the anti-thyroid antibodies [22,23,24,25].

When it comes to the metabolic interplay (particularly obesity and lipid profiles), a relationship with thyroid anomalies, such as hypothyroidism, was reported [26,27,28,29]. We found no correlation between the values of blood TSH levels and TgAbs; there was also no correlation between the values of FT4 and TgAbs or between the TPOAb and TgAb levels, while the blood levels of these antibodies were not associated with the serum total cholesterol or triglycerides, respectively, in the studied sub-groups. This might be explained by a relatively similar profile with concern to the thyroid panel (according to our assessments) in the AS versus AS-negative participants, and also by the fact that the subjects with clinically manifested severe hypothyroidism (myxedema) were not included. Obesity affected one-third of the patients (similar rates in AS and AS-negative individuals were confirmed) with statistically significant differences in the obesity grades (grade I affected 65% of the non-AS group as opposite to 30.4% in the AS group). Overt or subclinical hypothyroidism (regardless of the autoimmune cause) may interfere with certain weight gain or a resistance to losing weight, and this may be corrected via levothyroxine replacement [30,31,32]. Notably, in this study we excluded the patients with chronic respiratory or cardiovascular conditions (as potential complications of obesity) in order to restrict their influence in regard to the admission complaints. On the other hand, this is a dual perspective, and a higher rate of thyroid status impairment was reported in obese children and adults when compared to members of the population of an average weight [33,34,35,36]. For instance, an analysis published in 2023 by Bambini et al. [33] showed (across 79 studies) that 12% of obese pediatric subjects (with an average age of 10.9 years) had a thyroid ailment, while increased TSH levels was the most frequent finding upon thyroid profile assessment. In addition, high thyroid antibodies were more frequent in obese versus non-obese children (7% versus 3%, *p* < 0.001). In obese adults, 62.2% of them had a thyroid disease, and the most frequent type was overt hypothyroidism (29.9%) with a similar prevalence of HT (as diagnosed based on the elevated blood antibodies) in obese versus non-obese subjects (of 14–15%) [33]. A similar interplay with diabetes and insulin resistance was found in patients diagnosed with thyroid anomalies, including HT-related hypothyroidism [37,38,39,40,41], but this was out of the scope of the present study since we intended to exclude the bias that comes with abnormal glucose level-associated symptoms.

Currently, asthenia or chronic fatigue, a rather frequent complaint in daily multidisciplinary practice, involves a certain burden in regard to health care systems [42,43,44]. While it might affect the quality of life and the functional status, it is important to pinpoint various underlying conditions that bring about secondary complications (with or without other clinically or sub-clinically manifested multi-organ anomalies) [45,46,47]. Physical examination and questions that are addressed during anamnesis might provide some insights in order to perform targeted investigations. However, the use of the rating scales may be confusing since there are no standard approaches and no specific biomarkers; also, generally, asthenia/fatigue may be isolated or part of a complex clinical picture or a syndrome [42,43,44].

In terms of limitations of the study, we first mention its small sample size. The numerous exclusion criteria restricted the number of the enrolled patients, including the fact that we specifically took into account a large panel of endocrine conditions that might present with asthenia [48,49,50]. We also note the late COVID-19 pandemic and early post-COVID-19 era as having a potential influence over the clinical status caused by the coronavirus infection, long COVID-19 syndrome, and even the vaccination against the virus [51,52,53]. The post-acute infection syndrome (lasting for one to three months) has been reported in 7% to 40% of the patients who underwent COVID-19 infection [51]. A multitude of pathogenic traits were incriminated, but these data are still a matter of debate [54,55,56]. A transitory adrenal insufficiency, as well as hypothyroidism, has been taken into consideration in individuals who developed the syndrome, which, on the other hand, was reported to be common in novel autoimmunity-related ailments [57,58]. Of note, headache or “newly daily persistent headache” (that might turn into chronic daily headache) has also been described as part of the syndrome by some authors [59,60,61]. This may be associated with a bias in the interpretation of the data collection amid anamnesis in included subjects, with headache being described in 26.7% of the studied population and more frequently in the AS versus non-AS group (35.6% versus 18%, *p* = 0.03). Globally, long COVID stands for multiple clinical elements that overlap or mimic chronic fatigue syndrome/myalgic encephalomyelitis, burn out, fibromyalgia, and even irritable bowel syndrome [62]. This requires a meticulous semiology-based approach to begin with [63,64,65].

Secondarily, as already mentioned, we did not use a clinical rating scale for asthenia/headache during anamnesis; rather, we used self-reported complaints. The implementation of such scores amid real-life settings varies with the center, and we used a retrospective data collection from different hospitals (self-reported asthenia). Of important note, a daily endocrine practice concerning the thyroid field does not include, as mandatory, the use of such scale, as opposed to other disorders seen in the rheumatologic field, for instance. The retrospective study design allowed for a real-world data analysis despite the bias that comes with collecting self-reported symptoms. Notably, no standard fatigue scale is generally recommended for HT patients at this point.

Thirdly, we did not apply TI-RADS (Thyroid Imaging, Reporting, and Data System) to collect the ultrasound data [66,67,68], and this was limited to the distinct registration results in each department. We restricted the confirmation of HT based on the abnormal serum antibodies levels as generally used in everyday clinical practice and not based on suggestive ultrasound features for a thyroid autoimmune disease. Fourthly, no particular screening protocols were applied to exclude a concurrent malignancy, while the adrenal insufficiency dynamic testing was based on a personalized decision in a selected sub-group (and not routinely performed in each patient). Of note, 10% of the AS group and 14% of the non-AS group had a non-HT autoimmune condition. Moreover, we identified that up to 20% of the studied population had a prior diagnosis of osteoporosis. We did not use any screening protocol for fracture risk assessment in high-risk individuals; however, we should mention that, particularly in older seniors, a low bone mineral density may be accompanied by frailty and sarcopenia that sometimes mimics chronic fatigue syndrome, an aspect that seemed to be exacerbated amid the recent COVID-19 pandemic [69,70,71,72].

Overall, a multifactorial panel of elements might contribute to asthenia in HT patients, including in those that presented a normal thyroid function, as, for instance, we mention the co-presence of a chronic inflammation of an autoimmune background (both at tissue and blood antibodies level) or of a heterogeneous spectrum of cardio-metabolic conditions. Additionally, in HT (also known as chronic lymphocytic thyroiditis), the abnormal modulation of an immune response might cause an over-expression of different immune-cell-derivate cytokines as contributors to an organ’s micro-environment, signal transduction pathway anomalies, and increased oxidative stress, as seen in other ailments such as infections or cancers, whereas asthenia is reported as a traditional, non-specific clinical symptom [73,74,75,76,77]. Overall, thyroid “health” stands for a major contributor to the general status of wellbeing, and anomalies such as HT with/without thyroid dysfunction might “spread their wings” to the clinical picture, which remains the starting point amid a multidisciplinary practice [78,79,80,81,82].

Current knowledge gaps include the fact that symptomatic HT patients (apart from the presence of a clinical hypothyroidism) might display a different combination of pathogenic elements versus the sub-group, which remains completely asymptomatic. Also, a multidisciplinary algorithm might help the integration of the thyroid profile assessment among other causes of asthenia. Future research directions might involve larger longitudinal cohorts to quantify the presence of asthenia in HT versus non-HT subjects, respectively, versus other (non-HT) autoimmune co-morbidities. Moreover, the usefulness of routinely applying fatigue scales in daily endocrine practice should be clarified.

## 5. Conclusions

Clinical presentation in HT remains heterogeneous, and we analyzed a specific frame in this panel, namely, asthenia as a presentation complaint in patients who were newly diagnosed with HT amid real-life settings across a multi-center, retrospective study. According to the study’s findings, the asthenia group was associated with a higher rate of hypothyroidism (as defined by a serum TSH > 4.5 μIU/mL), with it being notable that almost one-third of the patients had increased TSHs. This remains a useful clinical clue amid primary and secondary health care. Total cholesterol and triglycerides were associated with the patients’ ages, which is consistent with prior reported data. One out of three patients suffered from obesity, and one out of ten individuals had a second (non-HT) autoimmune disorder (other than adrenal insufficiency), as similarly found in the AS versus non-AS group. Overall, these data pinpointed the fact that, in daily practice, asthenia as an admission complaint is not a reliable indicator of an underlying thyroid dysfunction or a higher level of serum antibodies against the thyroid in patients without a full clinical picture of thyrotoxicosis or myxedema.

## Figures and Tables

**Figure 1 life-14-01380-f001:**
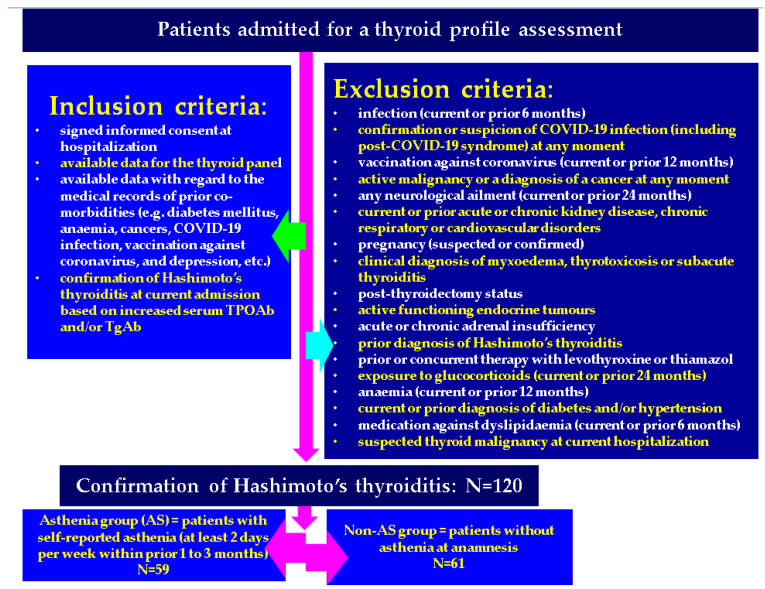
Flowchart of the inclusion and exclusion criteria for the studied population and studied sub-groups.

**Figure 2 life-14-01380-f002:**
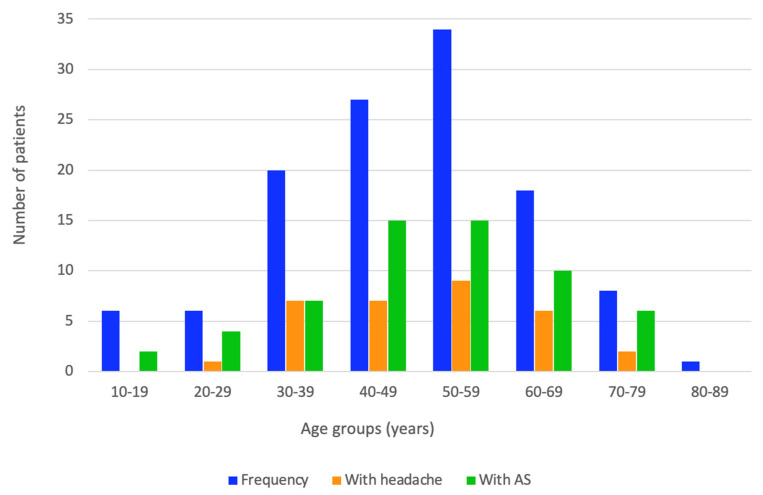
Bar chart of the patients’ number with self-reported headache and asthenia (AS) at admission within the age groups (this analysis is provided for the entire cohort of 120 patients diagnosed with HT).

**Figure 3 life-14-01380-f003:**
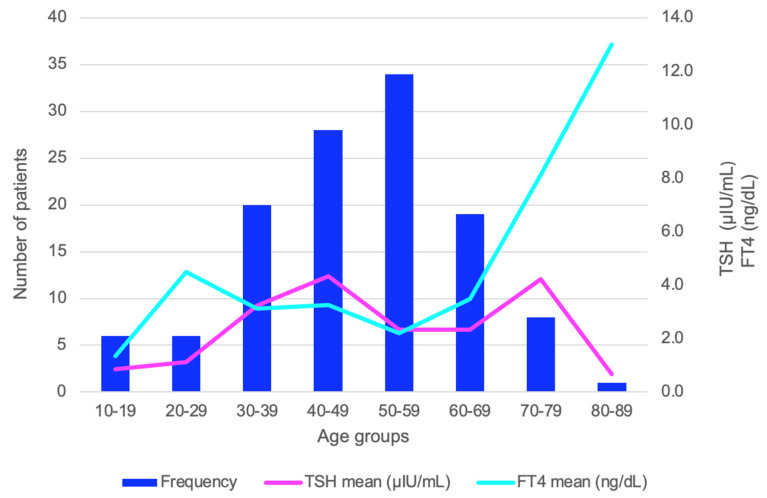
Bar and line chart of TSH and FT4 variation by age groups (the analysis is provided for the entire studied population of 120 patients diagnosed with HT aged between 10 and 81 years).

**Figure 4 life-14-01380-f004:**
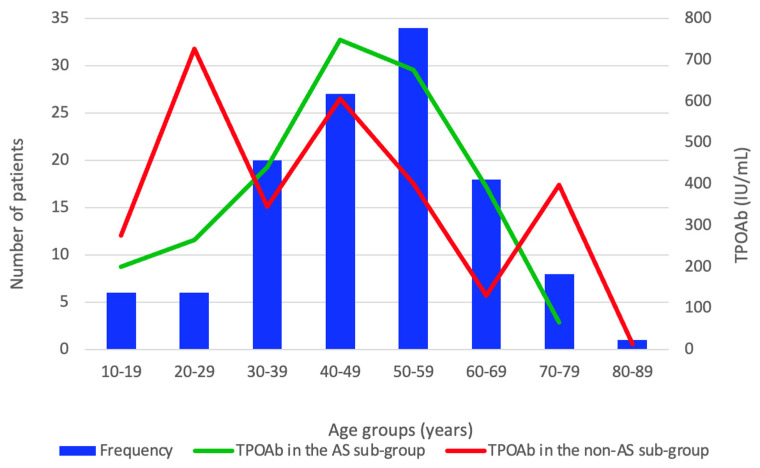
Bar and line chart of TPOAb distribution by age groups and the AS group (the age groups include the entire studied population of 120 subjects diagnosed with HT, while the AS group included 59 patients confirmed with HT and self-reported asthenia).

**Figure 5 life-14-01380-f005:**
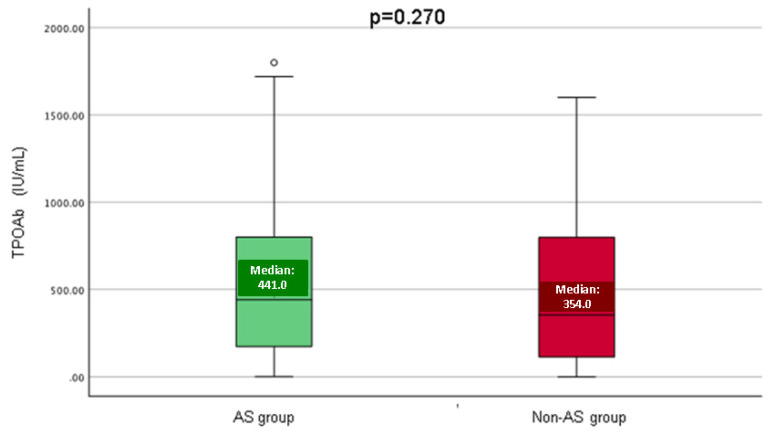
Boxplots showing TPOAbs in the AS versus non-AS groups (59 patients were included in AS group, representing 49.2%, with a mean age of 49.3 years, and 61 subjects were included in the non-AS group, with an average age of 47.1 years).

**Figure 6 life-14-01380-f006:**
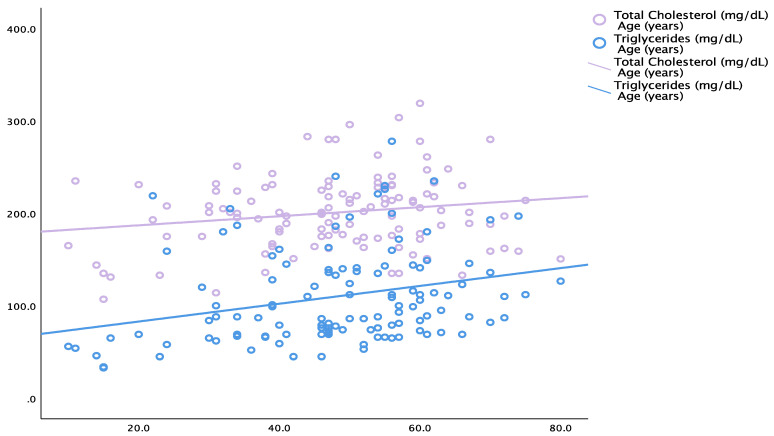
Correlation between the patients’ age and the level of blood total cholesterol, respectively, and between the patients’ age and the blood level of triglycerides (entire studied population, meaning AS and non-AS groups; N = 120; mean age of 48.2 ± 14.8 years).

**Figure 7 life-14-01380-f007:**
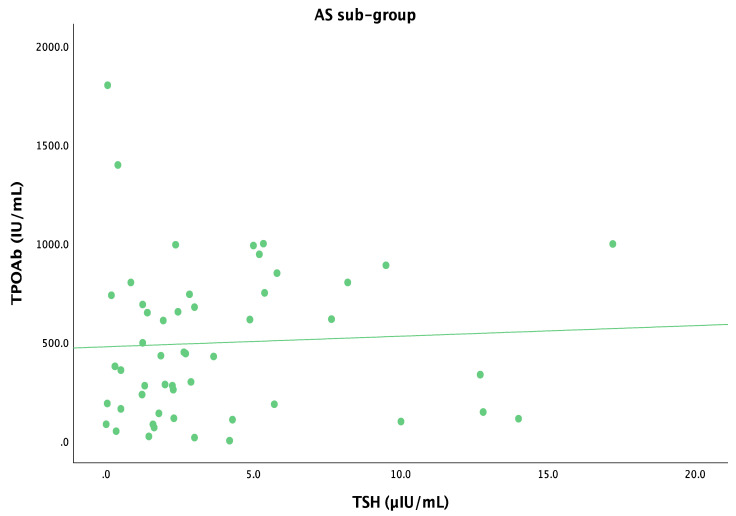
Scatterplot showing the relation between TSHs and TPOAbs in AS group (N = 59 patients with HT and asthenia, with a mean age of 49.3 ± 14.7 years).

**Figure 8 life-14-01380-f008:**
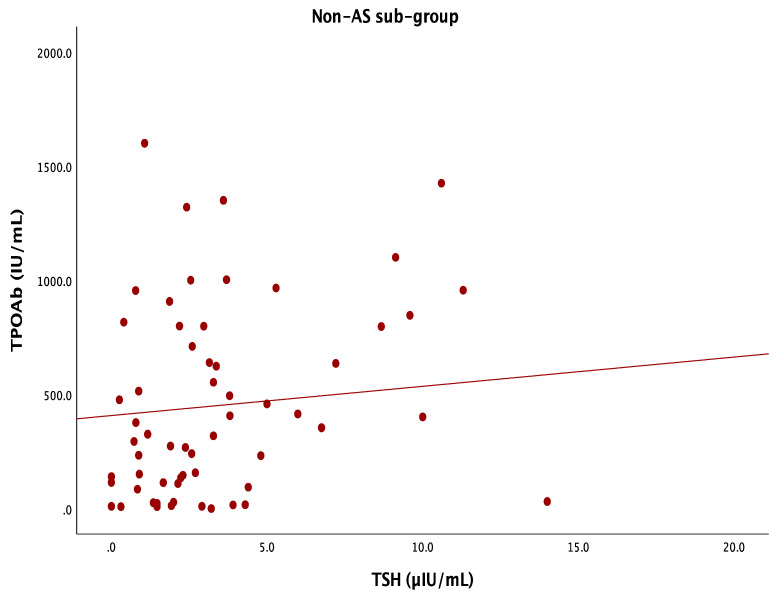
TSH-TPOAb correlation in the AS-negative group (N = 61 subjects with HT, with an average age of 47.1 ± 14.8 years).

**Table 1 life-14-01380-t001:** Demographic features, and co-morbidities in the studied population (N = 120) (data were generated with a student’s *t*-test for equality of means; a chi-square test was used for association between groups).

Parameter	Total Cohort (N = 120)	AS Group (N = 59)	AS-Negative Group (N = 61)	*p*-Value Between AS and AS-Negative Groups
Age (mean ± SD)	48.2 ± 14.8	49.3 ± 14.7	47.1 ± 14.8	0.426
Area of residence (rural, %)	33 (27.5)	25 (42.3)	8 (13.1)	**0.001**
Autoimmune diseases (number, %)	16 (13.3)	6 (10.1)	9 (14.8)	0.448
Obesity (number, %)	43 (35.8)	23 (39.0)	20 (32.8)	0.479
Osteoporosis (number, %)	27 (22.5)	15 (25.4)	12 (19.7)	0.451
Headache (number, %)	32 (26.7)	21 (35.6)	11 (18.0)	**0.030**

**Table 2 life-14-01380-t002:** Obesity grades analysis among the studied groups: the AS versus non-AS group (data were generated with a Fisher’s exact test for the association between the studied groups).

Obesity Grade	Total Cohort Number (% from Obese Patients)	AS Group Number (% from Obese Patients with AS)	AS-Negative Group Number (% from Obese Patients Without AS)	*p*-Value
Grade I	20 (46.5)	7 (30.4)	13 (65.0)	**0.035**
Grade II	15 (34.9)	9 (39.1)	6 (30.0)	
Grade III	8 (18.6)	7 (30.4)	1 (5.0)	

**Table 3 life-14-01380-t003:** Hypothyroidism rate (as defined by serum TSH > 4.5 μIU/mL) in the studied population.

Studied Population	Number of Patients (N) (Total Patients Within the Group)	Patients with Hypothyroidism (TSH > 4.5 μIU/mL) N (% from the Entire Sub-Group)
Entire studied cohort	120	37 (30.8)
AS group	59	23 (39.0)
AS-negative group	61	14 (23.0)
10–19 years	6	0 (0.0)
20–29 years	6	1 (16.7)
30–39 years	20	7 (35.0)
40–49 years	27	11 (40.7)
50–59 years	34	10 (29.4)
60–69 years	18	7 (38.9)
70–79 years	8	1 (12.5)
80–89 years	1	0 (0.0)

**Table 4 life-14-01380-t004:** Thyroid and lipids assays in the studied groups (Abbreviations: M = median, Q = quartile; data were generated with Mann–Whitney test).

Parameter	Total Cohort (N = 120)	AS Group (N = 59)	AS-Negative Group (N = 61)	*p*-Value	Normal Range
TSH (µIU/mL), M (Q1, Q3)	2.7 (1.4, 5.4)	2.8 (1.4, 8.2)	2.6 (1.3, 4.4)	0.275	0.4–4.5
FT4 (ng/dL), M (Q1, Q3)	1.2 (1.0, 1.6)	1.2 (1.0, 1.6)	1.3 (1.0, 1.5)	0.339	0.89–1.76
TPOAb (IU/mL), M (Q1, Q3)	403.5(136.0, 800.5)	441.0 (161.5, 801.0)	354.0 (111.9, 798.8)	0.270	0–50
TgAb (IU/mL), M (Q1, Q3)	57.5 (25.7, 364.5)	49.1 (25.0, 437.0)	69.0 (31.8, 257.5)	0.944	0–60
Total cholesterol (mg/dL), M (Q1, Q3)	201.0 (174.3, 228.0)	201.0 (176.0, 228)	201 (164.5, 228)	0.442	0–200
Triglycerides (mg/dL), M (Q1, Q3)	94.0 (69.0, 139.8)	111.0 (74.0, 141.0)	87.0 (67.0, 136.5)	0.234	0–150

**Table 5 life-14-01380-t005:** Correlation between the thyroid profile and patients’ age, respectively, as well as lipid profiles in the entire cohort (N = 120), AS group (N = 59), and non-AS group (N = 61) (data were generated based on Spearman’s correlation coefficient).

Parameter		Age (Years)	Total Cholesterol (mg/dL)	Triglycerides (mg/dL)
**Entire cohort (N = 120)**				
TSH (µIU/mL)	Correlation coefficient	−0.141	0.004	−0.078
	*p*-value	0.125	0.0967	0.399
FT4 (pmol/L)	Correlation coefficient	0.132	−0.086	0.010
	*p*-value	0.152	0.351	0.914
TPOAb (IU/mL)	Correlation coefficient	−0.086	0.141	−0.039
	*p*-value	0.348	0.125	0.671
TgAb (IU/mL)	Correlation coefficient	0.039	0.117	0.145
	*p*-value	0.672	0.202	0.113
**AS group (N = 59)**				
TSH (µIU/mL)	Correlation coefficient	−0.089	0.033	−0.029
	*p*-value	0.323	0.709	0.744
FT4 (pmol/L)	Correlation coefficient	0.098	−0.085	0.060
	*p*-value	0.282	0.349	0.508
TPOAb (IU/mL)	Correlation coefficient	−0.156	0.103	−0.066
	*p*-value	0.084	0.252	0.464
TgAb (IU/mL)	Correlation coefficient	0.035	0.069	0.185
	*p*-value	0.699	0.444	**0.040**
**AS-negative group (N = 61)**				
TSH (µIU/mL)	Correlation coefficient	−0.129	−0.012	−0.075
	*p*-value	0.147	0.896	0.397
FT4 (pmol/L)	Correlation coefficient	0.092	−0.039	0.001
	*p*-value	0.303	0.663	0.990
TPOAb (IU/mL)	Correlation coefficient	−0.009	0.096	−0.014
	*p*-value	0.921	0.273	0.876
TgAb (IU/mL)	Correlation coefficient	0.022	0.105	0.012
	*p*-value	0.808	0.234	0.891

**Table 6 life-14-01380-t006:** Correlation of the biochemical variables (lipid profile) and the patients’ age (Pearson’s correlation coefficient).

Parameter		Correlation Coefficient	*p*-Value
**Entire cohort (N = 120)**			
Total cholesterol	Age	**0.180**	**0.049**
	Triglycerides	**0.324**	**<0.001**
Triglycerides	Age	**0.277**	**0.002**
**AS group (N = 59)**			
Total cholesterol	Age	−0.107	0.236
	Triglycerides	0.104	0.249
Triglycerides	Age	0.125	0.169
**AS-negative group (N = 61)**			
Total cholesterol	Age	**0.246**	**0.006**
	Triglycerides	**0.319**	**<0.001**
Triglycerides	Age	**0.338**	**<0.001**

**Table 7 life-14-01380-t007:** Thyroid ultrasound features of the studied population (N = 120) (generated with Fisher’s exact test for association between groups).

Thyroid Ultrasound Trait	Total Studied Population (N = 120)	AS Group (N = 59)	AS-Negative Group (N = 61)	*p*-Value
Homogeneity				
Homogeneous, number (%)	8 (6.7)	3 (5.1)	5 (8.2)	0.494
Inhomogeneous, number (%)	112 (93.3)	56 (94.9)	56 (91.8)	
Echogenicity				
Hypoechoic, number (%)	85 (70.8)	41 (69.5)	44 (72.1)	0.575
Isoechoic, number (%)	31 (25.8)	15 (25.4)	16 (26.2)	
Hyperechoic, number (%)	4 (3.3)	3 (5.1)	1 (1.6)	
Nodules ≥5 mm diameter				
Present, number (%)	25 (20.8)	15 (25.4)	10 (16.4)	0.355
Absent, number (%)	95 (79.2)	44 (74.6)	51 (83.6)	

## Data Availability

The research data supporting this study’s findings are not publicly available. Further inquiries can be directed to the corresponding author.

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
