# Peer review of "A Real-Life Study in Patients Newly Diagnosed with Autoimmune Hashimoto’s Thyroiditis: Analysis of Asthenia as Admission Complaint"

_life, 2024, doi:10.3390/life14111380_

Round 1

Reviewer 1 Report

Comments and Suggestions for Authors

Manuscript ID: life-3254555

Manuscript title: A real-life study in patients newly diagnosed with autoimmune Hashimoto’s thyroiditis: analysis of asthenia as admission complaint

This manuscript investigates the prevalence and clinical significance of asthenia (fatigue) in patients newly diagnosed with Hashimoto’s thyroiditis (HT). The authors aim to determine whether asthenia correlates with thyroid dysfunction or relevant biochemical markers, using a real-world, multi-center, retrospective study design. While the topic is significant and addresses a common clinical issue in endocrinology, several major and minor concerns need to be addressed to strengthen the study’s overall impact and rigor.

Major:

- The abstract should be a single paragraph and follow the structured abstract style without headings. It is currently too long and includes excessive detail, particularly in the methods and results sections. Additionally, the abstract contains repetitive elements that mirror content from the main manuscript. It should be condensed to focus on key points, removing unnecessary statistical details.

- The introduction needs to be expanded. More information should be provided on HT, including its definition, epidemiology, symptoms, diagnostic approaches, and possible therapies. A more thorough background on asthenia and its relevance to HT should also be included. Additionally, in the section outlining the study’s aim, the authors should better justify the importance of investigating asthenia as a diagnostic marker and the choice of research methods.

- Sections 2.2 and 2.3: The inclusion and exclusion criteria should be consolidated in section 2.2 for clarity and to improve flow.

- The retrospective design is suitable for collecting real-world data but carries inherent biases, particularly when relying on self-reported symptoms such as asthenia. While the authors mention this limitation, it would be helpful to expand on how they managed retrospective data to minimize bias and confounding factors. Additionally, the absence of objective fatigue assessments (e.g., validated fatigue scales) could reduce the precision of the findings. The authors should clarify why these measures were not used and provide more detail on the limitations of relying solely on self-reported data.

- Please provide the consent number and the date of issuance of the ethical approval by the relevant bioethics committee (section 2.5.).

- While the results are presented clearly, a more in-depth analysis of the findings is needed. For instance, the study found no strong correlation between asthenia and thyroid dysfunction or antibody levels. The manuscript would benefit from a discussion of why these correlations were absent, particularly in light of existing literature. Could other factors (e.g., subclinical conditions or comorbidities) explain the lack of correlation? Expanding on this will enhance the study's scientific contribution.

- The study's conclusions should more thoroughly highlight the clinical implications of these findings. How can these results influence the diagnosis and management of HT in clinical practice? The authors should also address current knowledge gaps and suggest future research directions.

- The manuscript requires editorial revisions. Several sections are repetitive, and some figures/tables are unclear. The entire manuscript would benefit from a more careful review and reorganization to improve readability.

Minor:

- Keywords should be arranged in alphabetical order.

- Consider preparing a flowchart for section 2.2 to visually present the inclusion and exclusion criteria for the study.

- In Table 1, it would be helpful to explicitly state that the p-value refers to the comparison between the asthenia (AS) group and the non-AS group.

- Figure 2 duplicates the data presented in Table 2 and could be removed for conciseness.

- Increase the font size in Figure 5 for better readability.

- The data presented in Figure 1 should be described in more detail within the results section of the manuscript.

- Line 247: The correlation mentioned here is statistically insignificant and should be rephrased accordingly.

- Some figure legends could benefit from additional detail. The legends should provide enough information to understand the figures without referring back to the main text. More descriptive captions for all figures would make the results clearer for the reader.

- It would be interesting if the authors to hypothesize about the underlying mechanisms behind asthenia in HT patients, especially in those where thyroid function appears normal. While this study does not directly address mechanisms, discussing potential pathways (e.g., autoimmunity, inflammation) could provide a more comprehensive understanding of the symptom in HT.

Comments on the Quality of English Language

The manuscript’s language is generally clear but requires major revisions for grammatical accuracy, punctuation, and sentence flow to improve clarity and readability. Additionally, there are typographical errors that need correction.

Author Response

Response to Review 1 Comments

Dear Reviewer,

Thank you very much for your time and your effort to review our manuscript.

We are very grateful for providing your valuable feedback on the article.

Here is our response and related amendment that has been made in the manuscript according to your review (marked in yellow color).

Manuscript ID: life-3254555

Manuscript title: A real-life study in patients newly diagnosed with autoimmune Hashimoto’s thyroiditis: analysis of asthenia as admission complaint 

This manuscript investigates the prevalence and clinical significance of asthenia (fatigue) in patients newly diagnosed with Hashimoto’s thyroiditis (HT). The authors aim to determine whether asthenia correlates with thyroid dysfunction or relevant biochemical markers, using a real-world, multi-center, retrospective study design.

While the topic is significant and addresses a common clinical issue in endocrinology, several major and minor concerns need to be addressed to strengthen the study’s overall impact and rigor.

Thank you very much. We really appreciate it!

We addressed them point by point as follows:

Major:

The abstract should be a single paragraph and follow the structured abstract style without headings. It is currently too long and includes excessive detail, particularly in the methods and results sections. Additionally, the abstract contains repetitive elements that mirror content from the main manuscript. It should be condensed to focus on key points, removing unnecessary statistical details.

Thank you very much. According to your recommendation, we adjusted the abstract. Thank you

The introduction needs to be expanded. More information should be provided on HT, including its definition, epidemiology, symptoms, diagnostic approaches, and possible therapies. A more thorough background on asthenia and its relevance to HT should also be included. Additionally, in the section outlining the study’s aim, the authors should better justify the importance of investigating asthenia as a diagnostic marker and the choice of research methods.

Thank you very much. According to your recommendation, we expanded the Introduction section (regarding HT), the Objective micro-section, and provided more insights with respect to asthenia in HT, including at Discussion section. Thank you

Sections 2.2 and 2.3: The inclusion and exclusion criteria should be consolidated in section 2.2 for clarity and to improve flow.

Thank you very much. According to your recommendation, we merged them into a single sub-section (in addition to the study design section). Thank you

The retrospective design is suitable for collecting real-world data, but carries inherent biases, particularly when relying on self-reported symptoms such as asthenia. While the authors mention this limitation, it would be helpful to expand on how they managed retrospective data to minimize bias and confounding factors. Additionally, the absence of objective fatigue assessments (e.g., validated fatigue scales) could reduce the precision of the findings. The authors should clarify why these measures were not used and provide more detail on the limitations of relying solely on self-reported data.

Thank you very much. The bias of self-reported data was mentioned and discussed at Discussion section, including that lack of using standard scales. Thank you

Please provide the consent number and the date of issuance of the ethical approval by the relevant bioethics committee (section 2.5.).

Thank you very much. According to your recommendation, we added the information in the mentioned sub-section, but all these data are already provided at the end of the main text according to the MDPI rules. Thank you

While the results are presented clearly, a more in-depth analysis of the findings is needed. For instance, the study found no strong correlation between asthenia and thyroid dysfunction or antibody levels. The manuscript would benefit from a discussion of why these correlations were absent, particularly in light of existing literature. Could other factors (e.g., subclinical conditions or comorbidities) explain the lack of correlation? Expanding on this will enhance the study's scientific contribution.

Thank you very much. We expanded these data at Discussion. Thank you

The study's conclusions should more thoroughly highlight the clinical implications of these findings. How can these results influence the diagnosis and management of HT in clinical practice? The authors should also address current knowledge gaps and suggest future research directions.

Thank you very much. Based on your useful recommendation, we added at Discussion the current gaps and further research in HT/asthenia and revisited the Conclusion section. Thank you

The manuscript requires editorial revisions. Several sections are repetitive, and some figures/tables are unclear. The entire manuscript would benefit from a more careful review and reorganization to improve readability.

Thank you very much. We reorganized the text.

Minor:

Keywords should be arranged in alphabetical order.

Thank you very much. According to your recommendation, we corrected the key words. Thank you

Consider preparing a flowchart for section 2.2 to visually present the inclusion and exclusion criteria for the study.

Thank you very much. According to your recommendation, the flowchart was introduced in Methods section. Thank you for this interesting suggestion.

In Table 1, it would be helpful to explicitly state that the p-value refers to the comparison between the asthenia (AS) group and the non-AS group.

Thank you very much. We added this information within Table 1.

Figure 2 duplicates the data presented in Table 2 and could be removed for conciseness.

Thank you very much. We removed Figure 2 from the main text. We respectfully mention that we kept it as supplementary figure in order to make a clearer presentation of those data. Thank you

Increase the font size in Figure 5 for better readability.

Thank you very much. We corrected it.

The data presented in Figure 1 should be described in more detail within the results section of the manuscript.

Thank you very much. We provided more details with regard to the findings displayed in Figure 1.

Line 247: The correlation mentioned here is statistically insignificant and should be rephrased accordingly.

Thank you very much. We corrected it.

Some figure legends could benefit from additional detail. The legends should provide enough information to understand the figures without referring back to the main text. More descriptive captions for all figures would make the results clearer for the reader.

Thank you very much. We expanded the figures legends.

It would be interesting if the authors to hypothesize about the underlying mechanisms behind asthenia in HT patients, especially in those where thyroid function appears normal. While this study does not directly address mechanisms, discussing potential pathways (e.g., autoimmunity, inflammation) could provide a more comprehensive understanding of the symptom in HT.

Thank you very much. According to your recommendation, we extended the issue of potential pathogenic mechanisms or contributors to asthenia in studied population at Discussion section. Thank you

Comments on the Quality of English Language: The manuscript’s language is generally clear, but requires major revisions for grammatical accuracy, punctuation, and sentence flow to improve clarity and readability. Additionally, there are typographical errors that need correction.

Thank you very much. We revised the text and the English language. Thank you

Thank you very much.

Reviewer 2 Report

Comments and Suggestions for Authors

Oct 16, 2024

Dear Author,

1.     The manuscript, life-3254555, is within the scope of the journal.

2.     The section “Keywords” should be compatible with the MeSH database.

3.     The topic is an important issue in endocrinology and thyroidology. Therefore, we evaluate the author’s hypothesis as valuable. However, this issue might be discussed throughout the manuscript in different conditions. Therefore, the section “Discussion” must be revised, and the reference list must be enriched with some current and updated articles, such as “Revisiting femoral cartilage thickness in cases with Hashimoto's thyroiditis in thyroidology”, “Repercussion of thyroid dysfunctions in thyroidology on the reproductive system: Conditio sine qua non?”, “Programmed cell death-1 and its ligands: Current knowledge and possibilities in immunotherapy.”, and “World Thyroid Day 2023 in thyroidology: no overlook thyroid dis-eases to opt for “thyroid health” purposes.”

4.     The orthographical and grammatical errors have been recognized throughout the text. Of note, the manuscript must be revised by the authors. 

5. The manuscript includes some short paragraphs, subheadings, and essays. Therefore, they must be merged in the most proper way, meticulously. For instance, 1st and 2nd subheadings of the materials and methods might be conjoint.

6.     The manuscript might be accepted after major revisions are completed. 

Best Regards,

Reviewer, Life

Comments on the Quality of English Language

The orthographical and grammatical errors have been recognized throughout the text. Of note, the manuscript must be revised by the authors. 

Author Response

Response to Review 2 Comments

Dear Reviewer,

Thank you very much for your time and your effort to review our manuscript.

We are very grateful for your insightful comments and observations, also, for providing your valuable feedback on the article.

Here is a point-by-point response and related amendments that have been made in the manuscript according to your review (marked in yellow color).

Dear Author,

The manuscript, life-3254555, is within the scope of the journal.

Thank you very much. We really appreciate it!

The section “Keywords” should be compatible with the MeSH database.

Thank you very much. According to your recommendation we corrected the key words. Thank you

The topic is an important issue in endocrinology and thyroidology. Therefore, we evaluate the author’s hypothesis as valuable. However, this issue might be discussed throughout the manuscript in different conditions. Therefore, the section “Discussion” must be revised, and the reference list must be enriched with some current and updated articles, such as “Revisiting femoral cartilage thickness in cases with Hashimoto's thyroiditis in thyroidology”,  “Repercussion of thyroid dysfunctions in thyroidology on the reproductive system: Conditio sine qua non?”, “Programmed cell death-1 and its ligands: Current knowledge and possibilities in immunotherapy.”, and

“World Thyroid Day 2023 in thyroidology: no overlook thyroid dis-eases to opt for “thyroid health” purposes.”

Thank you very much. According to your recommendation we expanded the pathogenic insights and connected ailments in HT at Discussion section and followed those mentioned data. Thank you

The orthographical and grammatical errors have been recognized throughout the text. Of note, the manuscript must be revised by the authors. 

Thank you very much. We revised the text and the English language. Thank you

The manuscript includes some short paragraphs, subheadings, and essays. Therefore, they must be merged in the most proper way, meticulously. For instance, 1st and 2nd subheadings of the materials and methods might be conjoint.

Thank you very much. According to your recommendation we merged the mentioned subheadings. Thank you

The manuscript might be accepted after major revisions are completed. 

Thank you very much. We followed your recommendations point by point as shown. Thank you

Comments on the Quality of English Language: the orthographical and grammatical errors have been recognized throughout the text. Of note, the manuscript must be revised by the authors. 

Thank you very much. We revised the manuscript. Thank you

Thank you very much.

Reviewer 3 Report

Comments and Suggestions for Authors

Dear Claudiu Nistor and Oana-Claudia Sima

Thank you for your manuscript regarding a really complex issue

The relatively small number of patients alongside the effect of COVID infection are strong limitations of your study

On the other hand the design of your study is appropriate

Comments on the Quality of English Language

Moderate editing of English language required

Author Response

Response to Review 3 Comments

Dear Reviewer,

Thank you very much for your time and your effort to review our manuscript.

We are very grateful for your insightful comments and observations, also, for providing your valuable feedback on the article.

Here is a point-by-point response and related amendments that have been made in the manuscript according to your review (marked in yellow color).

Dear Claudiu Nistor and Oana-Claudia Sima,

Thank you for your manuscript regarding a really complex issue.

Thank you very much. We really appreciate it!

The relatively small number of patients alongside the effect of COVID infection are strong limitations of your study.

Thank you very much. These aspects have been addressed at Discussion section. Indeed, they are part of the real-world data collection amid recent medical and social reality with respect to the pandemic. Thank you

On the other hand the design of your study is appropriate.

Thank you very much.

Comments on the Quality of English Language: Moderate editing of English language required.

Thank you very much. We revised the text and the English language. Thank you

Thank you very much.

Round 2

Reviewer 1 Report

Comments and Suggestions for Authors

Manuscript ID: life-3254555

Manuscript Title: A Real-Life Study in Patients Newly Diagnosed with Autoimmune Hashimoto’s Thyroiditis: Analysis of Asthenia as an Admission Complaint

22/10/2024

Dear Authors,

Thank you for your diligent efforts in revising the manuscript and thoroughly addressing my previous comments. I appreciate the substantial improvements you have made, particularly in refining the abstract, expanding the introduction, and enhancing the clarity of the methods and results sections. Your responsiveness to the feedback has significantly strengthened the overall quality and readability of the manuscript.

In this next round of review, I would like to provide additional feedback to further refine the manuscript and ensure that the study’s contributions are clear and well-supported while maintaining a high standard of scientific rigor and presentation. Below, I have outlined some additional comments for your consideration:

- The manuscript still contains minor linguistic, grammatical, and punctuation errors that need attention.

- The resolution of the figures is low, making it difficult to read the data. I suggest exporting the graphics at a higher resolution to improve their quality.

Comments on the Quality of English Language

The manuscript still contains minor linguistic, grammatical, and punctuation errors that need attention.

Author Response

Response to Review 1 Comments – second round

Dear Reviewer,

Thank you very much for your time and your effort to review our manuscript.

We are very grateful for providing your valuable feedback on the article.

Here is our response and related amendment that has been made in the manuscript according to your review (marked in green color).

Manuscript ID: life-3254555

Manuscript Title: A Real-Life Study in Patients Newly Diagnosed with Autoimmune Hashimoto’s Thyroiditis: Analysis of Asthenia as an Admission Complaint

22/10/2024

Dear Authors,

Thank you for your diligent efforts in revising the manuscript and thoroughly addressing my previous comments. I appreciate the substantial improvements you have made, particularly in refining the abstract, expanding the introduction, and enhancing the clarity of the methods and results sections. Your responsiveness to the feedback has significantly strengthened the overall quality and readability of the manuscript.

Thank you very much.

In this next round of review, I would like to provide additional feedback to further refine the manuscript and ensure that the study’s contributions are clear and well-supported while maintaining a high standard of scientific rigor and presentation. Below, I have outlined some additional comments for your consideration:

The manuscript still contains minor linguistic, grammatical, and punctuation errors that need attention.

Thank you very much. We revised the text.

The resolution of the figures is low, making it difficult to read the data. I suggest exporting the graphics at a higher resolution to improve their quality.

Thank you very much. We improved the figures.

Comments on the Quality of English Language: The manuscript still contains minor linguistic, grammatical, and punctuation errors that need attention.

Thank you very much. We revised the text.

Thank you very much.

….

Reviewer 2 Report

Comments and Suggestions for Authors

Oct 20, 2024

Dear Author,

I am pleased to inform you that your manuscript, life-3254555.R1can be accepted for publication in the Life.

 Best Regards,

Reviewer, Life

Comments on the Quality of English Language

Minor editing of English language required.

Author Response

Response to Review 2 Comments – second round

Dear Reviewer,

Thank you very much for your time and your effort to review our manuscript.

We are very grateful for your insightful comments and observations, also, for providing your valuable feedback on the article.

Here is a point-by-point response and related amendments that have been made in the manuscript according to your review (marked in green color).

Dear Author,

I am pleased to inform you that your manuscript, life-3254555.R1, can be accepted for publication in the Life.

Best Regards,

Reviewer, Life

Thank you very much. We revised the text.

Comments on the Quality of English Language: Minor editing of English language required.

Thank you very much.
